# The effect of antenatal care on perinatal outcomes in Ethiopia: A systematic review and meta-analysis

Kasiye Shiferaw[1]*, Bizatu Mengiste[2], Tesfaye Gobena[3], Merga Dheresa[1]

**1** School of Nursing and Midwifery, College of Health and Medical Sciences, Haramaya University, Harar, Ethiopia, **2** St. Paul's Hospital Millennium Medical College, Addis Ababa, Ethiopia, **3** Department of Environmental Health Science, College of Health and Medical Sciences, Haramaya University, Harar, Ethiopia

These authors contributed equally to this work.
* sifkas.gem2@gmail.com

**Data Availability Statement:** All relevant data are within the manuscript and Supporting Information files.

**Funding:** The author(s) received no specific funding for this work.

## Abstract

### Background

The estimated annual global perinatal and neonatal death is four million. Stillbirths are almost equivalent to neonatal mortality, yet they have not received the same attention. Antenatal care is generally thought to be an effective method of improving pregnancy outcomes, but its effectiveness as a means of reducing perinatal mortality has not been evaluated in Ethiopia. Therefore, we will identify the pooled effect of antenatal care on perinatal outcomes in Ethiopia.

### Methods

Medline, Embase, Cinahl, African journal online and Google Scholar was searched for articles published in English language between January 1990 and May 2020. Two independent assessors selected studies and extracted data from eligible articles. The Risk of Bias Assessment tool for Non-Randomized Studies was used to assess the quality of each included study. Data analysis was performed using RevMan 5.3. Heterogeneity and publication bias were assessed using $I^2$ test statistical significance and Egger's test for small-study effects respectively. The random effect model was employed, and forest plot was used to present the risk ratio (RR) with 95% confidence interval (CI).

### Results

Thirteen out of seventeen included studies revealed antenatal care utilization had a significant association with perinatal outcomes. The pooled risk ratio by the random-effects model for perinatal death was 0.42 (95% CI: 0.34, 0.52); stillbirth 0.34 (95% CI: 0.25, 0.46); early neonatal death 0.85 (95% CI: 0.21. 3.49).

### Conclusion

Women who attended at least one antenatal care visit were more likely to give birth to an alive neonate that survives compared to their counterpart. Therefore, the Ethiopian Ministry

**Competing interests:** The authors have declared that no competing interests exist.

of health and other stakeholders should design tailored interventions to increase antenatal care utilization since it has been shown to reduce perinatal mortality.

## Introduction

Globally, an estimated four million perinatal and neonatal deaths occur annually [1–3]. In addition, an estimated 2.6 million babies were stillborn in 2015, only a 19% decrease since 2000. Ninety eight percent of stillbirth occurred in low and middle income countries (LMICs) and 77% of these occurred in the south Asia and Sub-Saharan Africa (SSA), thus showing little progress in SSA [2]. Majority of the stillbirths (60%) occurred during the antepartum period were mainly due to untreated maternal infection, hypertension, and poor fetal growth [2], which are preventable. The perinatal mortality rate across SSA was 35 per 1000 live births [4]. In Ethiopia, there are high proportions of stillbirths and early neonatal deaths [5,6], being one of the top ten countries with highest stillbirth numbers, and the high perinatal mortality rate (33 per 1000 live births) is coupled with high percent of low birth weight babies (13% of babies weighing less than 2500 grams at birth) [2,7,8].

The increase in perinatal mortality is more likely due to increased stillbirths and reduced antenatal visits [9]. ANC is a vital intervention for successful maternal and child health, globally [10]. Attending less than 50% of recommended or inadequate ANC visits was associated with adverse pregnancy outcomes [11–17]. Stillbirths are a reflection of ANC accessibility and utilization [18]. Women with no ANC had significantly increased risk of stillbirths [19]; mortality and morbidity of mothers and newborns was reduced for those who had optimal utilization of ANC services [20]. Furthermore, the risk of developing neonatal mortality was decreased for women who received as little as one ANC follow up [21–27], but the effect on perinatal outcomes is unknown.

Studies revealed that low birth weight (LBW) was associated with not attending at least five to eight ANC visits, not receiving any ANC during the first trimester and not having access to certain ANC contents [28–30]; LBW is a contributing factor to stillbirths [31]. However, there are conflicting results on the effectiveness of ANC interventions on maternal and newborn health outcomes [32–36]. There are inconsistencies in the studies regarding the benefits of ANC in reducing perinatal mortality [17,36–40]; studies revealed perinatal mortality was not affected by no and inadequate ANC [41]; other studies showed improved ANC did not reduce perinatal or neonatal mortality [42]. Benefits of ANC were reported by some but not all care programs regarding perinatal mortality [43]; however, ANC has not been compellingly shown to improve birth outcomes [44]. Furthermore, the focused ANC model is associated with more perinatal deaths than models comprised of at least eight ANC contacts [45].

Reduction in an availability and utilization's gaps of ANC practice is needed to end preventable deaths of newborns [46]. Failure to improve birth outcomes by 2035 will result in an estimated 116 million deaths, 99 million survivors with disability [47], and an additional 52 million stillbirths [47,48]. There are no pooled estimates of the effect of ANC on perinatal outcomes in Ethiopia; therefore, we aimed to assess the effect of ANC on perinatal outcomes in Ethiopia in this systematic review and meta-analysis.

## Methods

The Preferred Reporting Items for Systematic Reviews and Meta-Analysis (PRISMA) checklist was used in the preparation of the systematic review methodology [49]. The systematic review

was registered on the PROSPERO prospective register of systematic reviews after piloting the study selection process (registration number PROSPERO 2020: CRD42020188340).

## Eligibility criteria

Assessment for eligibility was conducted and studies were included in this review if (i) the study involved a delivering/laboring women or newborn babies or women of child-bearing age or pregnant women or postpartum women; (ii) the study reported the outcomes (perinatal death, stillbirth, early neonatal death); (iii) the ANC utilization was considered as factors/exposure for the outcomes; (iv) the study was done in the perinatal period and the author(s) defined perinatal outcomes (perinatal mortality) as death of newborn between 28 weeks' of gestation and seven days postpartum; (v) it was an observational study design (cross-sectional, case-control or cohort study design) and (v) English language article.

We excluded studies from the review that focused only on the number of ANC visits based on full-text assessment.

**PICO. Population:** Newborn after 28 weeks' gestation and survived seven days postpartum.

**Intervention:** utilized at least one ANC visit.

**Comparison:** Newborns whose mothers received at least one ANC service as compared to newborns whose mothers did not.

**Outcome:** Newborn death during perinatal period (from 28 weeks' of gestation to 7 days postpartum).

## Information sources and search strategy

Medline (via PubMed), EMBASE, and CINAHL were searched for (S1–S3 Tables) articles published in the English language between January 1990 and June 2020, using the keywords "antenatal care", "prenatal care", "maternity care", "maternal health care", "obstetrics", "maternal health services", "pregnancy care", "perinatal mortality", "perinatal death", "early neonatal mortality", "early neonatal death" "stillbirth", "newborn mortality", "newborn death", "perinatal outcomes", "fetal death" "infant death", "infant mortality" AND "Ethiopia". Moreover; we thorough literatures search was performed on Google Scholar and African Journal Online (AJOL). A search combining MeSH and key terms connecting population, intervention and outcomes of interest was performed.

## Study selection

The study selection involved several steps. First, the title and abstract were selected independently by the review authors using the inclusion criteria. Second, after removing the duplicates, the full reports of all titles that met the inclusion criteria were independently identified by review authors. Third, the review authors screened the full text reports to decide whether the studies meet the eligibility criteria. Finally, any disagreements among review authors were resolved through discussion or review authors who did not participate in step one thru three decided whether to include or exclude the article. An attempt was made to meet study authors for additional information by email and in order to have put reasons for excluding studies (Fig 1).

## Data extraction

Each studies' relevance was checked based on their topic, objectives and methodology. Two independent reviewers (KS & BM) completed and verified the data extraction, using a standardized form with explicit inclusion and exclusion criteria. If not resolved by discussion of reviewers, the

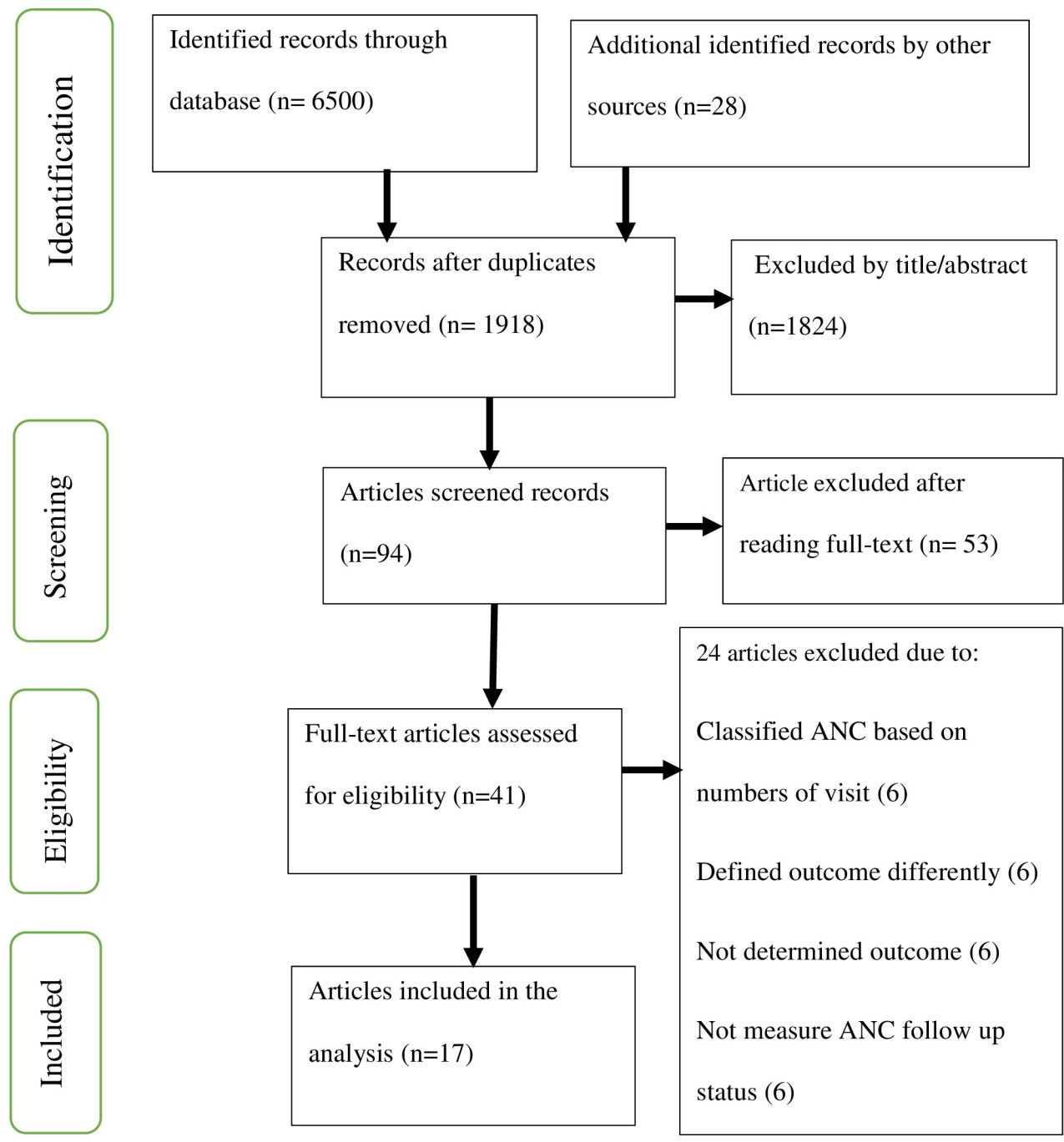

**Fig 1. Flowchart of search results for included studies.**

third or fourth author was consulted to decide on the disagreement. For each study, the first author's last name, publication year, design, setting, sample size, study period, sample age, the definition of outcomes, population, outcome and comparison groups were documented.

In this review, our evaluation of perinatal outcomes related to the death of the newborn from 28 weeks' of gestation to seven days postpartum (i.e., fetal death, stillbirth, and early

neonatal death) were reported [50–52]. Antenatal care 'no ANC visit at all' or 'one or more ANC visit' were the two classifications of the exposure variable. An attempt to contact study authors to request information, such as missing data, was made, if necessary and unfortunately there was no study which was excluded do to missing data.

## Data items

**Antenatal care** is defined as 'a woman having one or more health facility visits for a pregnancy check-up by a skilled attendant during her pregnancy' [12,23,53–58].

Focused ANC model is four visits providing essential evidence based interventions–a package to achieve the full life-saving potential that ANC promises for women and babies [59].

Perinatal outcomes/mortality refers to the number of stillbirths and deaths in the first week of life [23,42,60,61]. In this review, perinatal death (as study authors defined) or reviewers added the number of stillbirths and early neonatal deaths or available outcomes between stillbirths and early neonatal deaths to estimate overall perinatal outcomes/mortality rate.

Stillbirth was defined as fetal deaths after 28 weeks of gestation [5,62,63].

Skilled attendant refers to a midwife, doctor or nurse who has been educated, trained and accredited to manage normal pregnancies, childbirth and the immediate postnatal period and identify, manage and/or refer women and newborns with complications [64].

Early neonatal mortality was defined as neonatal deaths in the first week of life after being delivered in the age of viability (28 weeks of gestation and above) [5,56].

## Individual study's risk of bias

The review authors assessed all selected studies rigorously for inclusion in the review. The Risk of Bias Assessment Tool for Non-Randomized Studies (RoBANS) [65] was used to assess the quality of each included study. Studies were evaluated across six groups (selection bias, attrition bias, detection bias, performance bias, confounding bias and reporting bias). Each domain was assigned one of three possible groups for each of the involved studies: 'low risk', 'high risk' and 'unclear'. RoBANS is shown in S4 Table.

## Synthesis and analysis of data

Statistical analysis was carried out in RevMan version 5.3. A DerSimonian and Laird random effects model [66] was used to measure ANC's overall effect on perinatal mortality and the risk ratio was measured with a 95% confidence interval. We calculated the $I^2$ statistic which describes the percentage of total variation among studies to assess heterogeneity among studies. An $I^2$ statistical value of 25%, 50% and 75% representing low, moderate and high heterogeneity respectively [67]. A p-value less than 0.05 was considered as statistically significant both for risk ratio and heterogeneity.

Sensitivity analysis was conducted to assess the stability of results and test individual study effects on the meta-analysis using leave one out method. Furthermore, possible sources of heterogeneity were explored using subgroup analysis. Egger's test for small-study effects was used to investigate potential publication bias (p-value > 0.1) [68].

## Results

### Search results

The initial search identified 1918 unique citations. Of these, 1824 and 53 were excluded on title/abstract alone and following full-text review respectively. Furthermore, 6 articles classified ANC based on number of visits [69–74], 6 articles defined their outcomes differently [62,75–

79], 6 articles did not determine outcome at all [26,53,80–83] and 6 articles had no ANC fol-low-up status [84–89] and therefore they all were excluded. Lastly, 17 articles were retained for final review (Fig 1).

## Characteristics of studies

The review included studies from all regions in Ethiopia; the majority were from Amhara and Oromia. Nine cross-sectional, six case-control and two cohort studies were included in the meta-analysis. The sample size of the studies ranged from 300 to 12560. Among the included studies, 5 and 12 were community-based and facility-based, respectively. A total of 51729 study samples were included, of which 2951 newborns died during the perinatal period, mak-ing the perinatal mortality rate 41 per 1000 total births (total deliveries, total stillbirths and total early neonatal deaths), excluding case-control studies in which total numbers of live births at the time of the study were unknown. Similarly, the stillbirth rate and early neonatal mortality rate were 38 per 1000 total births (stillbirths and live births) and 19 per 1000 live births. Table 1 displays the characteristics of the 17 included primary studies.

## Individual study's risk of bias

The risk of bias assessment for all included studies is shown in Table 2. The risk of bias in selection of participants into the study was low for all studies. The bias due to missing or incomplete data was low in most of the studies, although a few studies have unclear explana-tion. The performance bias during measurement of exposure variable was low in fourteen and unclear in three studies. However, the risk of detection bias was high in all studies. The risk of confounding bias was low in thirteen, high in three and unclear in one study. The bias due to reporting of results was low in fifteen and unclear in two studies. See S4 Table.

## Pooled effect size of ANC on perinatal outcomes

Among the seventeen studies included in the analysis, thirteen studies with at least one ANC visit showed statistically significant associations with perinatal outcomes, whereas four studies had no statistically significant association. Similarly, the pooled effect size for perinatal death by the random-effect model was 0.42 (95% CI: 0.34, 0.52) for babies born to women who received at least one ANC follow-up as compared to newborns whose mothers did not receive any ANC follow-up (Fig 2). Furthermore, the pooled stillbirth and early neonatal death effect size by random effects model was 0.34 (95% CI: 0.25, 0.46) and 0.85 (95% CI: 0.21. 3.49) respectively.

## Heterogeneity of the studies

There was overall substantial heterogeneity across studies ($I^2$ = 87%, p-value < 0.001), as well as within subgroups for sample size, design and place. Heterogeneity that was present in the overall meta-analysis was partially explained with stratification by study design and place. For example, in a subgroup analysis, cohort studies' (RR = 0.83[95% CI: 0.67–1.02]; p-value = 0.45 for heterogeneity test, $I^2$ = 0%) and community-based studies (RR = 0.64[95% CI: 0.51–0.80]; p-value = 0.23 for heterogeneity test, $I^2$ = 29%) were not statistically heterogeneous (p-value > 0.10); however, heterogeneity was present when the subgroup analysis was performed by sample size (Table 3).

Sensitivity analysis was performed for the outcome variable to observe a significant change in risk ratio and confidence interval. The meta-analysis resulted in no substantial difference in the overall risk ratio during the sequential removal of each study from the analysis. For

**Table 1. Characteristics of studies revealing the effect of ANC on perinatal outcomes in Ethiopia.**

| No | Authors | Design | Study setting | Study period | Sample size | Population | Sample age | Outcomes variable | Operational definition | ANC status | Perinatal outcomes | |
|---|---|---|---|---|---|---|---|---|---|---|---|---|
| | | | | | | | | | | | Yes | No |
| 1 | Adane etal. 2014 [90] | Cross-sectional study | Facility-based | February 2013 | 481 | Laboring women | <20, 20–35, 35+ | Stillbirth | Stillbirth was defined as the birth of an infant that has died in the womb or during intra-partum after 28 weeks of gestation. | Yes | 18 | 397 |
| | | | | | | | | | | No | 16 | 50 |
| 2 | Goba et al. 2017 [91] | Case-control study | Facility-based study | From February 1 to September 30, 2016 | 378 | Delivering women | <24, 25–34, 35+ | Perinatal death | Patients who experienced stillbirth or early neonatal death were classified as the case group and those whose neonates survived until discharge or for at least 7 days were control group. | 0 visits | 19 | 9 |
| | | | | | | | | | | 1–3 visits | 89 | 149 |
| | | | | | | | | | | ≥4 visits | 18 | 94 |
| 3 | Roro et al. 2018 [92] | Nested case-control study | Community-based study | Between March 2011 to December 2012 | 4438 | Newborn babies | 15–19, 20–24, 25–29, 30–34, 35+ | Perinatal mortality | Perinatal death is defined as the sum of stillbirth and early neonatal death. | Yes | 56 | 121 |
| | | | | | | | | | | No | 17 | 25 |
| 4 | Welegebriel et al. 2017 [93] | Case-control study | Facility-based study | From January 2011 to 2015 | 540 | Mothers registered in for maternal health service utilization | <20, 20–34, 35+ | Stillbirth | Not defined | Yes | 69 | 278 |
| | | | | | | | | | | No | 66 | 127 |
| 5 | Worede and Dagnew 2019 [94] | Unmatched case-control | Facility-based study | From 1st January to 30th April 2019 | 420 | Delivering women | <20, 20–34, 35+ | Stillbirth | Case is defined as fetal death after 28 weeks of pregnancy (either pre-partum or intra-partum stillbirth) | Yes | 47 | 284 |
| | | | | | | | | | | No | 37 | 52 |
| 6 | Getiye and Fantahun 2017 [95] | Unmatched case-control study | Facility-based study | From January 1/ 2014 to Dec 31/ 2014 | 1113 | Delivering women | 15–19, 20–24, 25–29, 30–34, 35+ | Perinatal outcome | Perinatal mortality is total number of deaths in the perinatal period | Yes | 354 | 724 |
| | | | | | | | | | | No | 22 | 13 |
| 7 | Tilahun & Assefa 2017 [96] | Cross-sectional study | Facility-based | Not specified | 413 | Delivering women | <20, 20–34, 35–45 | Stillbirth | Not defined | Yes | 17 | 321 |
| | | | | | | | | | | No | 16 | 59 |
| 8 | Berhie and Gebresilassie 2016 [97] | Cross-sectional study | Community-based study | From September 2010 through June 2011 | 12,560 | Women of child-bearing age | 15–24, 25–34, 35+ | Stillbirth | Pregnancy losses occurring after seven completed months of gestation are defined as stillbirths. | No ANC visit | 273 | 3828 |
| | | | | | | | | | | Visited at least once | 118 | 3172 |

(*Continued*)

**Table 1.** (Continued)

| No | Authors | Design | Study setting | Study period | Sample size | Population | Sample age | Outcomes variable | Operational definition | ANC status | Perinatal outcomes | |
|---|---|---|---|---|---|---|---|---|---|---|---|---|
| | | | | | | | | | | | Yes | No |
| 9 | Tilahun and Gaym 2008 [98] | Case-control study | Facility-based | From May 15, 2006 to August 15, 2006 | 390 | Delivering mothers | <20, 20–34, 35+ | Perinatal Mortality | Perinatal mortality (case) were mothers with a singleton pregnancy who were admitted to the labor ward and had a stillbirth or suffered an early neonatal death after delivery. | Unbooked | 43 | 14 |
| | | | | | | | | | | Booked | 87 | 246 |
| 10 | Ballard et al. 2016 [99] | Cross-sectional study | Community-based | Between May and December 2014 | 4442 | Women of child-bearing age | Not mentioned | Stillbirth | The stillbirth was delivering a dead neonate after a pregnancy lasting 7 months or more. | Received ANC | 42 | 2437 |
| | | | | | | | | | | Not received ANC | 53 | 1921 |
| 11 | Eyob and Worku 2003 [100] | Cross-sectional study | Facility-based | From l^st January 1995 to December 31, 1996 | 8986 | Delivering mothers | Not mentioned | Perinatal death | Not defined | Unbooked | 283 | 1770 |
| | | | | | | | | | | Booked | 301 | 6632 |
| 12 | Worku et al. 2013 [14] | Prospective cohort study | Community-based | From December 1, 2011 to August 31, 2012 | 727 | Pregnant women | <20, 20–34, 35+ | Perinatal death | Definition taken from WHO guideline monitoring emergency obstetric care | Yes | 13 | 240 |
| | | | | | | | | | | No | 23 | 451 |
| 13 | Lakew et al. 2017 [101] | Cross-sectional | Community-based | 2014 | 2555 | Women of child-bearing age | <24, 25–34, 35+ | Stillbirth | Stillbirth outcomes was characterized as the introduction of a newborn child that has passed on in the womb or amid intra-partum following 28 weeks of growth | No ANC visit | 9 | 138 |
| | | | | | | | | | | ANC 1 + visit | 7 | 231 |
| 14 | Berhan 2014 [102] | Retrospective cohort study | Facility-based | Between January 2006 and December 2011 | 9619 | Women that gave birth | <20, 20–34, 35+ | Perinatal death | Perinatal status defined the fetal or early neonatal survival (from 28 weeks of pregnancy age up to the first 7 days of newborn age) | Yes | 124 | 283 |
| | | | | | | | | | | No | 90 | 149 |
| 15 | Chekol A., 2011 [103] | Cross-sectional | Facility-based | From September 2008 to August 2009 | 581 | Laboring women | 15–19, 20–29, 30–42 | Perinatal death | It is fetal death starting from 28 weeks of gestational age and the death of new born in the first week of life, which comprises late fetal and early neonatal deaths. | No | 36 | 93 |
| | | | | | | | | | | Yes | 33 | 419 |
| 16 | Aragaw Y., 2016 [104] | Cross-sectional | Facility-based | From September 11, 2012 to 10, 2013 | 3786 | Newborn babies | <20, 20–34, 35+ | Perinatal death | Not defined | Yes | 204 | 2765 |
| | | | | | | | | | | No | 169 | 648 |

(*Continued*)

**Table 1.** (Continued)

| No | Authors | Design | Study setting | Study period | Sample size | Population | Sample age | Outcomes variable | Operational definition | ANC status | Perinatal outcomes | |
|---|---|---|---|---|---|---|---|---|---|---|---|---|
| | | | | | | | | | | | Yes | No |
| 17 | Mihiretu A. et al, 2017 [105] | Cross-sectional | Facility-based | July, 2015 | 300 | Mothers who gave birth | <18, 18–34, 35+ | Perinatal death | Not defined | Yes | 10 | 142 |
| | | | | | | | | | | No | 42 | 107 |

instance, when a statistically insignificant study [14] and those study with wide confidence interval were excluded [101], the risk ratio of the effect of ANC did not change significantly or are within the confidence interval of pooled effect of ANC (0.32, 0.52).

An Egger's test for small-study effects showed no publication bias (p-value = 0.49). Therefore, there was no significant threat to the validity of the review.

## Discussion

The purpose of this review was to evaluate the effectiveness of focused ANC as a means of reducing perinatal mortality among women (pregnant, delivering, postpartum, and mothers) in Ethiopia. Seventeen eligible primary studies were identified evaluating ANC with a range of populations including pregnant women, laboring women and postpartum mothers and their perinatal outcomes. Literature throughout Ethiopia support the benefits of ANC's that provided by skilled attendants for the health of newborns. To improve ANC's effectiveness, numerous approaches and strategies have been employed in LMICs [106–108]. The focused ANC approach, developed in the 1990s by WHO has been implemented by most LMICs including Ethiopia [109,110].

The perinatal mortality and stillbirth rate were 41 and 38 per 1000 total births respectively in this meta-analysis which were slightly higher than the perinatal mortality rate in SSA (34.7 per 1000 total births) [4] however, lower than the pooled perinatal mortality rate (51.3 per

**Table 2. Individual studies risk of bias on effect of ANC on perinatal outcomes in Ethiopia.**

| Studies | Selection bias | Attrition bias | Performance bias | Detection bias | Confounding bias | Reporting bias |
|---|---|---|---|---|---|---|
| Adane et al. 2014 | Low | Low | Low | High | Low | Low |
| Ballard et al. 2016 | Low | Low | Low | High | High | Low |
| Berhan 2014 | Low | Low | Unclear | High | High | Low |
| Berhie and Gebresilassie 2016 | Low | Low | Low | High | Low | Low |
| Eyob and Worku 2003 | Low | Low | Unclear | High | High | Low |
| Getiye and Fantahun 2017 | Low | Low | Low | High | Low | Low |
| Goba et al. 2017 | Low | Low | Unclear | High | Low | Low |
| Lakew et al. 2017 | Low | Low | Low | High | Low | Low |
| Roro et al. 2018 | Low | Unclear | Low | High | Low | Low |
| Tilahun & Assefa 2017 | Low | Low | Low | High | Low | Low |
| Tilahun and Gaym 2008 | Low | Unclear | Low | High | Low | Low |
| Welegebriel et al. 2017 | Low | Unclear | Low | High | Low | Low |
| Worede and Dagnew 2019 | Low | Low | Low | High | Low | Low |
| Worku et al. 2013 | Low | Low | Low | High | Unclear | Low |
| Chekol A., 2011 | Low | Low | Low | High | Low | Low |
| Aragaw Y., 2016 | Low | Low | Low | High | Low | Unclear |
| Mihiretu A. et al, 2017 | Low | Low | Low | High | Low | Unclear |

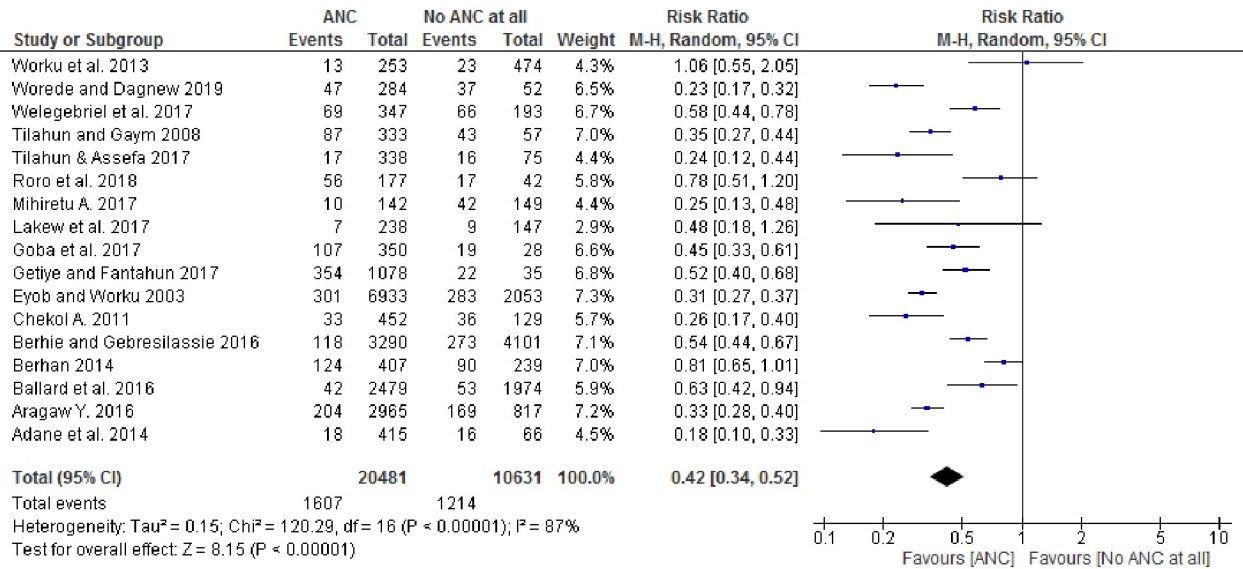

**Fig 2. Forest plot showing pooled effect of ANC on the perinatal outcomes in Ethiopia.**

1000 total births) and slightly higher than stillbirth rate in Ethiopia (37 per 1000 total births) [6]. The review in SSA utilized only demographic health survey data whereas the pooled perinatal mortality in Ethiopia included both demographic health survey and study data. The difference may be attributed to not only a variation in the study nature, sample size, and setting but also maternal and child health utilization and access to quality maternal and newborn health services [6]. However, early neonatal mortality rate was 19 per 1000 live births in this review which was lower than systematic reviews found in Ethiopia (30 per 1000 live births).

A global multipartner movement to end preventable maternal and newborn deaths and stillbirths, setting a target for national stillbirths less than 12 per 1000 live births and will reduce death and disability continuously, ensuring no newborn is left behind in all countries by 2030 [2,19]; however, this review, along with the EDHS [5] and another systematic review in Ethiopia [6] revealed that the perinatal mortality has remained stable for two decades. Using this study's perinatal mortality rate as a benchmark, the annual rate of reduction (ARR) must increase to achieve The Every Newborn Action Plan.

**Table 3. Studies' subgroup analysis modifying the effect of ANC on perinatal outcomes in Ethiopia.**

| Subgroup | Random effects RR(95% CI) | I-squared, p-value |
|---|---|---|
| Study design | | |
| Cross-sectional study | 0.34(0.27–0.44) | 78%, p-value < 0.001 |
| Case-control study | 0.45(0.33–0.61) | 85%, p-value < 0.001 |
| Cohort study | 0.83(0.67–1.02) | 0%, p-value = 0.45 |
| Study place | | |
| Community-based study | 0.64(0.51–0.80) | 29%, p-value = 0.23 |
| Facility-based study | 0.36(0.28–0.46) | 88%, p-value < 0.001 |
| Sample size | | |
| < 500 | 0.29(0.22–0.38) | 67%, p-value = 0.009 |
| 500–1000 | 0.53(0.26–1.05) | 86%, p-value = 0.0006 |
| > 1000 | 0.52(0.39–0.69) | 90%, p-value < 0.001 |

Pregnant women's attendance of at least one ANC follow-up had a statistically significant effect on perinatal mortality. This study found a 58% and 66% lower risk of perinatal mortality and stillbirth among women who attended at least one ANC by a skilled attendant in Ethiopia. The basic finding of this study was even limited ANC (as little as one visit) leads to better newborn outcomes compared with no ANC, and encouraging pregnant women to seek ANC would significantly impact perinatal mortality rate (PMR) and would be an important strategy to incorporate in planning initiatives aimed at reducing PMR; this appears to be consistent with studies from another countries [56,111]. The finding was also in line with the global network's population-based birth registry results in Africa, India, Pakistan and Guatemala [19]. A review in Asia also revealed a protective effect on perinatal mortality for women who used ANC and health facility delivery [39].

Antenatal care utilization and delivery at a health facility by a skilled attendant [112] who provides quality care are established as an intervention to reduce perinatal mortality [113–115]. This may be due to the women receiving interventions during her pregnancy, [116–118] which have a positive effect on lowering mortality; ANC also has an indirect impact since those women attending ANC are more likely to have a skilled birth attendant [39,112,119,120] hence, their newborns have access to basic neonatal resuscitation [121,122] which prevent perinatal mortality. Therefore, receiving high quality and an accessible health care services to reduce perinatal mortality is critical for pregnant women [123]. Skilled training of health care providers and resources of local primary healthcare facilities should be strengthened [124].

The factors associated with perinatal mortality (preterm labor, hypertensive disorders of pregnancy, intrauterine growth restriction, gestational diabetes) can be identified in the prenatal period, thus reinforcing the need to upgrade the continuum of care from initiation of ANC to complication management at health facilities [113,125].

A comprehensive database search was conducted to include all pertinent studies, and subgroup analysis was conducted to determine whether any specific study level factor described the outcomes. The large sample size of the analysis, could detect the effect of ANC on perinatal outcomes since the review included all studies conducted in Ethiopia. As a limitation, the systematic review and meta-analysis were based on English language and observational studies associated with inherent biases. We were unable to pool the overall effect of ANC for those studies that were based on the number of visits, since they did not define zero visits and therefore that were excluded. The study authors defined stillbirth and early neonatal death based on gestational age and the days of life of the newborn. The future research should focus on visits and specific ANC interventions that may affect perinatal outcomes.

## Conclusion

This review showed that women who received at least one ANC follow-up by a skilled attendant were less likely to experience perinatal mortality than those who did not. Thus, increasing a woman's ANC utilization by a skilled attendant is mandatory in Ethiopia to reduce perinatal mortality. Furthermore, to address perinatal mortality in the country, strategies should focus on women's mobilization to seek ANC services and facility-based deliveries.

## Supporting information

**S1 Table. Searching using Medline via PubMed.**
(DOCX)

**S2 Table. Searching using EMBASE (via Ovid).**
(DOCX)

**S3 Table. Searching using CINAHL.**
(DOCX)

**S4 Table. Assessment of risk of bias for individual study (RoBANS).**
(DOCX)

**S1 File. Completed PRISMA checklist.**
(DOC)

## Acknowledgments

We would like to acknowledge Haramaya University for providing a scholarship and stipend. We also thank Tara Wilfong for her constructive comments and language edition.

## Author Contributions

**Conceptualization:** Kasiye Shiferaw.

**Data curation:** Kasiye Shiferaw.

**Formal analysis:** Kasiye Shiferaw.

**Investigation:** Kasiye Shiferaw.

**Methodology:** Kasiye Shiferaw, Bizatu Mengiste, Tesfaye Gobena, Merga Dheresa.

**Software:** Kasiye Shiferaw.

**Supervision:** Bizatu Mengiste, Tesfaye Gobena, Merga Dheresa.

**Validation:** Kasiye Shiferaw, Bizatu Mengiste, Tesfaye Gobena, Merga Dheresa.

**Visualization:** Kasiye Shiferaw.

**Writing – original draft:** Kasiye Shiferaw.

**Writing – review & editing:** Kasiye Shiferaw, Bizatu Mengiste, Tesfaye Gobena, Merga Dheresa.

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
