## [Decision Letter · Decision Letter 0]

14 Sep 2020

PONE-D-20-22748

The effect of prenatal care on perinatal outcomes in Ethiopia: A systematic review and meta-analysis

PLOS ONE

Dear Dr. Kasiye Shiferaw,

Thank you for submitting your manuscript to PLOS ONE. After careful consideration, we feel that it has merit but does not fully meet PLOS ONE’s publication criteria as it currently stands. Therefore, we invite you to submit a revised version of the manuscript that addresses the points raised during the review process.

We look forward to receiving your revised manuscript.

Kind regards,

Georg M. Schmölzer

Academic Editor

PLOS ONE

Journal Requirements:

3. We noticed you have some minor occurrence of overlapping text with the following previous publications, which needs to be addressed:

https://journals.plos.org/plosone/article?id=10.1371%2Fjournal.pone.0222566

https://bmcpregnancychildbirth.biomedcentral.com/articles/10.1186/s12884-020-02880-5

https://nam.edu/wp-content/uploads/2016/09/Beyond-Survival-The-Case-for-Investing-in-Young-Children-Globally.pdf

In your revision ensure you cite all your sources (including your own works), and quote or rephrase any duplicated text outside the methods section. Further consideration is dependent on these concerns being addressed.

Reviewers' comments:

Reviewer's Responses to Questions

**Comments to the Author**

1. Is the manuscript technically sound, and do the data support the conclusions?

Reviewer #1: Yes

Reviewer #2: Yes

Reviewer #3: Yes

2. Has the statistical analysis been performed appropriately and rigorously? 

Reviewer #1: Yes

Reviewer #2: No

Reviewer #3: I Don't Know

3. Have the authors made all data underlying the findings in their manuscript fully available?

Reviewer #1: Yes

Reviewer #2: Yes

Reviewer #3: Yes

4. Is the manuscript presented in an intelligible fashion and written in standard English?

Reviewer #1: No

Reviewer #2: Yes

Reviewer #3: Yes

5. Review Comments to the Author

Reviewer #1: Comments to the Authors:

The authors of this manuscript present a meta-analysis of nationally representative publications from Jan 1990 – June 2020, that reported on the effect of prenatal care on perinatal outcomes in Ethiopia. The study addresses a gap in knowledge in the stillbirth and early neonatal death rates in Ethiopia, which is not always reported in low and middle-income countries, but has important implications for overall reduction of the neonatal mortality rate (NMR), and provides a practical path for policymakers as they implement strategies to mitigate conditions that contribute to preventable morbidity and mortality in that country.

General comments:

1)The study was thoughtfully designed and executed. However, throughout the manuscript, there is need for correction of grammar, spelling and syntax. Changes in other areas of this manuscript will inform changes in the discussion section.

2) The basic premise of this study is to show that even limited prenatal care (as little as one visit) leads to better neonatal outcomes compared with no prenatal care, and encouraging pregnant women to seek prenatal care would significantly impact neonatal mortality rate (NMR) and would be an important strategy to incorporate in planning initiatives aimed at reducing NMR; this appears to be consistent with the experience of similar countries.

3) Since it appears that the authors had access to granularity of data, it would have added more weight to the manuscript if they also reported on causes of neonatal death and/or maternal risk factors contributing to stillbirths and early neonatal death (ENND) in this population. This is important for health providers and policymakers as it gives information about the burden of maternal/neonatal morbidity as they prioritize resources and develop targeted public health policies to optimize maternal and neonatal survival. Furthermore, the information is relevant not only for the current pregnancy, but also future pregnancies, since it informs the need for heightened surveillance for at-risk mothers as causative factors contributing to stillbirths and ENND may recur in a later pregnancy.

4) Definitions need to be more explicit, e.g. “stillbirths were identified according to WHO definition as fetal loss at or after 28 weeks gestation”, which would qualify as late and term stillbirths according to U.S. definition: early stillbirth – loss between 20-27 completed weeks of pregnancy; late stillbirth – loss between 28-36 completed weeks of pregnancy; term stillbirth – loss at 37 or more completed weeks of pregnancy.

5) For consistency – choose prenatal care (PNC) or antenatal care (ANC) throughout the manuscript.

Specific comments:

1. In the Abstract section: Lines 10 and 28 need to be clarified

2. In the Introduction section: Some of the content would be best addressed in the Discussion section. Also, please clarify lines 28-30.

3. In the Methods section: Please explain why “newborns after 28 weeks gestation” was chosen as the study population – is it consistency of WHO definition of stillbirth, accepted age of viability in Ethiopia, consistency with analyzed studies included in the meta-analyses, or some other reason?

4. In the Methods section: Authors should clarify inclusion criteria, especially item (i).

5. In the Methods section: Overall, the methodology is acceptable – authors were correct in their modelling by using random effects models for meta-analysis and sensitivity analysis in search of robustness of results. Study design, study population and sample size were well-described, however, sample age among/within the various studies of would have added weight to the manuscript.

6. In the Methods section: Operational definition: firstly, authors should indicate that the WHO stillbirth definition is being used. Secondly, in discussing skilled attendants and their diploma – what type/level of diploma did they achieve. Third, early neonatal death is sometimes defined as neonatal death in those born >1000g, occurring in the first 7 days of life. However, there is no consistent definition with regards to weight and gestational age cutoffs, only consistency is death in the first 7 days of life.

7. In the Results section: “Characteristics of included studies” – please explain why case control studies were excluded from determination of perinatal mortality rate. In addition, there appears to be confusion regarding definitions. If perinatal mortality refers to “stillbirths and death in the first week of life”, how can perinatal mortality rate be given as 39 per 1000 live births, or are the authors referring to NMR?

8. In the Results section: Despite substantial heterogeneity, the pooled effect size by the random effect model and tight confidence intervals were reassuring and speaks to the robustness of results.

9. In the Results section: Authors state that “visual observation of the funnel plot summary showed no publication bias”. On average, the ability to visually discern publication bias from funnel plots is poor; and the plot included with this manuscript does not represent a symmetric inverted funnel shape, but rather an asymmetric funnel that may suggest fundamental difference between studies of higher and lower precision, which appears to be consistent with the presented risk of bias assessments. The funnel plot does not substantially contribute to the manuscript and should be excluded.

10. In the Discussion section: “developing countries”, “low and middle-income countries” is preferred terminology.

11. In the Discussion section: Lines 22-28 – please clarify…is the author trying to indicate that the annual rate of reduction (ARR) needs to be much higher than the current ARR in order to achieve The Every Newborn Action Plan goal by 2030.

12. As mentioned earlier, there are sentences in the introduction, that would be better placed in the discussion.

Reviewer #2: This is a great paper that outlines the effects of antenatal care on newborn outcomes and it has a potential to highlight important predictors of newborn outcomes

First the paper states that they are looking into pregnancy outcomes - however as we read the paper we come to learn that the authors are interested in in newborns who have survived at least 7 days of life. But surprisingly the authors also have stillbirth as one of the outcomes and a stillbirth and neonatal death are not the same thing. It is also important to specify that the pregnancy outcomes of interest at neonatal outcomes right from the title

Methods - please review the prisma P 2015 version to see how you can reorganize the subheadings in the methods sections for easy flow

It was not clear in the selection criteria who did the selection and how it was done

As part of the search strategy submitting a table that gives the detailed search in the different databases would make it easy for those replicating the search to come up with the exact same search that you have - this can be submitted as a supplemental file

For the inclusion criteria it was not clear what study designs would be considered although when we move to risk bias analysis the authors mention non-randomized studies. However, it is great to see that two people conducted the data extraction and risk bias assessment

The operational definitions can be part of the introduction, however, it would be important to keep the ones that are relevant for the focus of the paper

When it came to the results it was difficult to understand which outcome was being reported it would be great to analyze the different outcomes separately - also conduct a qualitative synthesis to find out what the authors mean by perinatal death (as this will include both stillbirth and neonatal death). Analyze neonatal death and stillbirth separately and if you are including stillbirth as an outcome you may have revise the population in your PICO question

It was great to see the Prisma flow diagram as it helped understand the selection process

Reviewer #3: Overall, the manuscript was well organized, adding to existing literature. The authors did a good job at providing sufficient background information on the issue at hand, however I would advise to provide more clarity when providing worldwide statistics on the effectiveness of prenatal care versus those for Ethiopia, thus allowing for the discussion and conclusions of the manuscript to have a stronger impact on its readers. I feel the main title of the manuscript and then those for the tables and figures could have been stronger, though current titles do make sense. I do not have the appropriate knowledge base to determine whether the methods and statistical analyses used were most accurate for this type of research, however, the data provided in the Methods and Results sections matched the tables and figures provided. I caution authors to proofread carefully, paying close attention to capitalization and punctuation.

6. PLOS authors have the option to publish the peer review history of their article (what does this mean?). If published, this will include your full peer review and any attached files.

Reviewer #1: **Yes: **Janine Y Khan

Reviewer #2: **Yes: **Kaboni Whitney Gondwe

Reviewer #3: No

---

## [Author Response · Author response to Decision Letter 0]

16 Oct 2020

Journal Requirements:

 Response- we have seen both guidelines and the manuscript was reorganized according these guidelines in the current version. 

Response- The native language speaker has seen the document very well. The address of the person who saw the manuscript is: 

“Tara Wilfong MD, MPH

Associate Professor, College of Health and Medical Sciences

Haramaya University

Fulbright Alumni, Ethiopia

099 363 3861

twdoc@ufl.edu; tara.wilfong@fulbrightmail.org”

The corrected manuscript was uploaded as recommended according to journal requirements. 

3. We noticed you have some minor occurrence of overlapping text with the following previous publications, which needs to be addressed:

https://journals.plos.org/plosone/article?id=10.1371%2Fjournal.pone.0222566

https://bmcpregnancychildbirth.biomedcentral.com/articles/10.1186/s12884-020-02880-5

https://nam.edu/wp-content/uploads/2016/09/Beyond-Survival-The-Case-for-Investing-in-Young-Children-Globally.pdf

In your revision ensure you cite all your sources (including your own works), and quote or rephrase any duplicated text outside the methods section. Further consideration is dependent on these concerns being addressed.

 Response- Thank you for the comments. The overlapping were checked in the current version and there were no overlapping. We tried to synthesize the idea as much as possible. We also used online free plagiarism checker to see if there is overlapping. We hope the problems in the previous version do not exist in the current version. Furthermore, it is our pleasure to correct if any such problems exist in the current version. 

Response- The comment is accepted and incorporated in the current version. 

Response to Reviewers

Reviewer #1: Comments to the Authors:

The authors of this manuscript present a meta-analysis of nationally representative publications from Jan 1990 – June 2020, that reported on the effect of prenatal care on perinatal outcomes in Ethiopia. The study addresses a gap in knowledge in the stillbirth and early neonatal death rates in Ethiopia, which is not always reported in low and middle-income countries, but has important implications for overall reduction of the neonatal mortality rate (NMR), and provides a practical path for policymakers as they implement strategies to mitigate conditions that contribute to preventable morbidity and mortality in that country.

General comments:

1)The study was thoughtfully designed and executed. However, throughout the manuscript, there is need for correction of grammar, spelling and syntax. Changes in other areas of this manuscript will inform changes in the discussion section.

Response- the comment is constructive and important for the clarity of the manuscript, hence the native language speaker has edited the manuscript. The grammar, spelling and syntax correction were made thoroughly in the manuscript. 

2) The basic premise of this study is to show that even limited prenatal care (as little as one visit) leads to better neonatal outcomes compared with no prenatal care, and encouraging pregnant women to seek prenatal care would significantly impact neonatal mortality rate (NMR) and would be an important strategy to incorporate in planning initiatives aimed at reducing NMR; this appears to be consistent with the experience of similar countries.

Response- Thank you for understanding our manuscript very well. We incorporated the comment you suggested to the manuscript as it exactly explain the finding. 

3) Since it appears that the authors had access to granularity of data, it would have added more weight to the manuscript if they also reported on causes of neonatal death and/or maternal risk factors contributing to stillbirths and early neonatal death (ENND) in this population. This is important for health providers and policymakers as it gives information about the burden of maternal/neonatal morbidity as they prioritize resources and develop targeted public health policies to optimize maternal and neonatal survival. Furthermore, the information is relevant not only for the current pregnancy, but also future pregnancies, since it informs the need for heightened surveillance for at-risk mothers as causative factors contributing to stillbirths and ENND may recur in a later pregnancy.

Response- You raised very important idea. As you mentioned, there are maternal, fetal and other factors that are contributing for perinatal deaths. Researchers did review ((Aminu et al. 2014 systematic literature review in low and middle income countries), (Reinebrant et al. 2017 systematic review globally) and (Berhan Y, and Berhan A, 2014 and Gedefaw et al. 2020 systematic review and meta-analysis in Ethiopia)) which identified very important determinant variables of pregnancy outcomes. These review have some gaps that we will address in the future review. But, the causative factors contributing to perinatal death (i.e. stillbirth and/or early neonatal death) covers vast concept to cover here and we plan to assess this part in another review. In this systematic review and meta-analysis, we need to know the effect of antenatal care on perinatal outcomes. 

4) Definitions need to be more explicit, e.g. “stillbirths were identified according to WHO definition as fetal loss at or after 28 weeks gestation”, which would qualify as late and term stillbirths according to U.S. definition: early stillbirth – loss between 20-27 completed weeks of pregnancy; late stillbirth – loss between 28-36 completed weeks of pregnancy; term stillbirth – loss at 37 or more completed weeks of pregnancy.

Response- Thank you for the comment. In this review we took operational definition of study authors which is conceding with WHO definition. The study authors didn’t classify the stillbirth based on gestational age, hence we couldn’t identify these stillbirth as early, late or term, since we relied on study authors’ definition. As far as, the study authors used WHO definition (fetal loss at or after 28 weeks gestation), it may qualify late or term stillbirth. 

5) For consistency – choose prenatal care (PNC) or antenatal care (ANC) throughout the manuscript.

Response- The comment is accepted; consistency very important and we chose the term ‘antenatal care’ throughout the manuscript instead of ‘prenatal care’. 

Specific comments:

1. In the Abstract section: Lines 10 and 28 need to be clarified

Response- thank you. The comment is incorporated in the manuscript, we hope it is now clear in the current version. 

2. In the Introduction section: Some of the content would be best addressed in the Discussion section. Also, please clarify lines 28-30.

Response- The comment is accepted, we moved one sentence ‘A global multipartner movement to end preventable maternal and newborn deaths and stillbirths, set a target for national stillbirth less than 12 per 1000 live births in all countries by 2030 [2, 10]’ to discussion section. As we go through the introduction there were no other sentences which we found important if moved to discussion section. We believe that introduction section sound well if these sentences retained. Furthermore, we reorganized the introduction section into four paragraphs for more clarity and precision. 

3. In the Methods section: Please explain why “newborns after 28 weeks gestation” was chosen as the study population – is it consistency of WHO definition of stillbirth, accepted age of viability in Ethiopia, consistency with analyzed studies included in the meta-analyses, or some other reason?

Response- Thank you for this important question. The World Health Organization (WHO) has defined stillbirth as ‘fetal death late in pregnancy’, deferring the gestational age (GA) when a miscarriage becomes a stillbirth to country policy. Sometimes stillborn babies are not weighed, in these cases a gestational age of 28 completed weeks or a body length of 35 cm can be taken as equivalent to 1000 gram birth weight. As a result, Ethiopia adopted this definition and defined stillbirth as “fetal deaths after 28 weeks of gestation” which have being practiced in national guideline. Furthermore, 28 weeks of gestations is age of fetal viability in Ethiopia. The researchers also used this definition of stillbirth in Ethiopia. It very important if WHO definition of stillbirth that included gestational age and weight of the newborn were used, but in Ethiopia weight of the stillbirth baby is not measured routinely. As a result, we put this problems in our limitation as ‘the study authors’ definition were only based on gestational age of the newborn.’ in the discussion section.

4. In the Methods section: Authors should clarify inclusion criteria, especially item (i).

Response- Thank you. As study authors mentioned, the studies were done on different population i.e. delivering/laboring women or newborn babies or women of child-bearing age or pregnant women or postpartum women (Table 1). Therefore, all the studies that have assessed the outcomes and exposure of interest were included in this analysis. We rewrote the inclusion criteria (i) as ‘the study involved a delivering/laboring women or newborn babies or women of child-bearing age or pregnant women or postpartum women’. We have also made clear other inclusion criteria as recommended in the current version of the manuscript. 

5. In the Methods section: Overall, the methodology is acceptable – authors were correct in their modelling by using random effects models for meta-analysis and sensitivity analysis in search of robustness of results. Study design, study population and sample size were well-described, however, sample age among/within the various studies of would have added weight to the manuscript.

Response- Thank you we have added ‘sample age’ to the manuscript as recommended. 

6. In the Methods section: Operational definition: firstly, authors should indicate that the WHO stillbirth definition is being used. 

Response- The comment is accepted. The definition was taken from WHO stillbirth definition in previous citations (already adopted from WHO definition). As we already mentioned above, Ethiopia adopted WHO definition which defined stillbirth as “fetal deaths after 28 weeks of gestation”. Furthermore, we cited WHO reference (from whom the definitions was adopted) in the current version of the manuscript. 

Secondly, in discussing skilled attendants and their diploma – what type/level of diploma did they achieve. 

Response- Thank you for the comments. We have substituted the previous definition with “Skilled attendant refers to a midwife, doctor, or nurse who been educated, trained and accredited to manage normal pregnancies, childbirth and an immediate postnatal period and identify, manage and/or refer women and newborns with complications” which is WHO definition of skilled attendant in the current version. The previous definition was taken from literature (Debelew et al. 2014 which defined skilled attendant as ‘Those who have trained to the level of Diploma and above was categorized as ‘‘skilled attendants’’). 

Third, early neonatal death is sometimes defined as neonatal death in those born >1000g, occurring in the first 7 days of life. However, there is no consistent definition with regards to weight and gestational age cutoffs, only consistency is death in the first 7 days of life.

Response- Yes, we share your concern. As you already mentioned, the articles we included in this review defined early neonatal death as ‘death of newborn in the first 7 days of life’ regardless of the weight of the newborn. It would have been better if weight of the newborn in addition to age in days were used for the definition. We reviewed all the included studies whether they considered weight as criteria for definition as recommended. But, the articles were not mentioned the weight and gestational age cutoffs in their definitions of the early neonatal death. Furthermore, there were no studies that we excluded because of including or excluding weight >1000g in their definitions. 

7. In the Results section: “Characteristics of included studies” – please explain why case control studies were excluded from determination of perinatal mortality rate. In addition, there appears to be confusion regarding definitions. If perinatal mortality refers to “stillbirths and death in the first week of life”, how can perinatal mortality rate be given as 39 per 1000 live births, or are the authors referring to NMR?

Response- We found your comment very important. We calculated stillbirth rate and early neonatal death rate separately (i.e. stillbirth rate, early neonatal mortality rate). Moreover, we calculated perinatal mortality rate adding stillbirth and early neonatal death to have overall estimation of perinatal mortality rate considering total live births (sample size in our case) as denominators. Most of the time case-control study design take case and control (the denominator is unknown) as a result it is not conducive to talk about magnitude or prevalence or rate of the disease. Similarly, previous review excluded case-control studies from rate calculation (e.g. Jena B.H., Biks G.A., Gelaye K.A. & Gete Y.K., 2020). However, the case-control study design that have total population (i.e. Roro et al. 2018 used nested case-control study design and the denominator is known) was included in the rate calculation. 

We added this sentence to current version of manuscript “In this review, we took perinatal death (as study authors defined) or we added the numbers of stillbirths and early neonatal deaths (perinatal death during perinatal period) or we took available outcome between of stillbirths and early neonatal deaths to estimate overall perinatal outcomes/mortality rate.”

8. In the Results section: Despite substantial heterogeneity, the pooled effect size by the random effect model and tight confidence intervals were reassuring and speaks to the robustness of results.

Response- Thank you. We tried to subgroup the studies by study design, sample size and setting and the heterogeneity explained by study design and setting. But there were no significant change of heterogeneity of the analysis during subgrouping by sample size. 

9. In the Results section: Authors state that “visual observation of the funnel plot summary showed no publication bias”. On average, the ability to visually discern publication bias from funnel plots is poor; and the plot included with this manuscript does not represent a symmetric inverted funnel shape, but rather an asymmetric funnel that may suggest fundamental difference between studies of higher and lower precision, which appears to be consistent with the presented risk of bias assessments. The funnel plot does not substantially contribute to the manuscript and should be excluded.

Response- Thank you. We share your concern “the ability to visually discern publication bias from funnel plots is poor”. As you suggested, we substituted the funnel plot with statistical method of assessment of publication bias (Egger's test for small-study effects). The Egger's test for small-study effects showed no statistical significance (H0=there is no small-study effect, H alternative=There is small-study effects, p-value was 0.49, therefore we failed to reject the H0), hence no publication bias in the manuscript. This finding match our funnel plot report of no publication bias. 

10. In the Discussion section: “developing countries”, “low and middle-income countries” is preferred terminology.

Response- Thanking you for the comment, we substituted ‘developing countries’ with ‘low and middle-income countries’ throughout the manuscript in the current version. 

11. In the Discussion section: Lines 22-28 – please clarify…is the author trying to indicate that the annual rate of reduction (ARR) needs to be much higher than the current ARR in order to achieve The Every Newborn Action Plan goal by 2030.

Response- Yes, thank you for the reconstruction of the sentence. we corrected the sentence as ‘however, this review, the EDHS [5] and review in Ethiopia [6] revealed that perinatal mortality remained stable in about two decade and considering our perinatal mortality rate as benchmark, the annual rate of reduction (ARR) needs to be much higher than the current ARR in order to achieve The Every Newborn Action Plan goal by 2030.’ in the current version of manuscript. 

12. As mentioned earlier, there are sentences in the introduction that would be better placed in the discussion.

Response- We brought sentence ‘A global multipartner movement to end preventable maternal and newborn deaths and stillbirths, set a target for national stillbirth less than 12 per 1000 live births in all countries by 2030 [2, 10]’ to discussion section. We prefer the other sentences retained in the introduction section to better show the gaps. 

Reviewer #2: This is a great paper that outlines the effects of antenatal care on newborn outcomes and it has a potential to highlight important predictors of newborn outcomes

First the paper states that they are looking into pregnancy outcomes - however as we read the paper we come to learn that the authors are interested in in newborns who have survived at least 7 days of life. But surprisingly the authors also have stillbirth as one of the outcomes and a stillbirth and neonatal death are not the same thing. It is also important to specify that the pregnancy outcomes of interest at neonatal outcomes right from the title

Response- Thank you for this credible idea, we share your concern “a stillbirth and neonatal death are not the same thing” and we defined perinatal outcomes as ‘In this review, we report our evaluation of perinatal outcomes related to death of newborn from 28 weeks’ of gestation to seven days postpartum (i.e. stillbirths and/or early neonatal deaths)’. Stillbirth was defined as fetal deaths after 28 weeks of gestation and early neonatal mortality was neonatal deaths in the first week of life after being delivered in the age of viability (28 weeks of gestation and above). 

These definition was based on authors’ definition which is conceding with world health organization of these terms. We did the analysis separately for effect of ANC on stillbirth and early neonatal death as well. Ten studies saw effect of ANC on stillbirth which have showed statistically significant association. But, only two study were saw effect of ANC on early neonatal death and have no statistically significant association. Therefore, perinatal outcomes/mortality in our case is the stillbirth and/or early neonatal death to have overall estimates. The following sentence included in the current version of manuscript after your recommendation “In this review, we took perinatal death (as study authors defined) or we added the numbers of stillbirths and early neonatal deaths or we took available outcome between of stillbirths and early neonatal deaths to estimate overall pooled perinatal outcomes/mortality rate.”

Methods - please review the prisma P 2015 version to see how you can reorganize the subheadings in the methods sections for easy flow

Response- Okay. We reorganize it in the current version.

It was not clear in the selection criteria who did the selection and how it was done

As part of the search strategy submitting a table that gives the detailed search in the different databases would make it easy for those replicating the search to come up with the exact same search that you have - this can be submitted as a supplemental file

Response- Okay, the search strategies for EMBASE, and CINAHL were added to previous search strategy of PubMed that was submitted as additional fines (S1-S3 Table). 

For the inclusion criteria it was not clear what study designs would be considered although when we move to risk bias analysis the authors mention non-randomized studies. However, it is great to see that two people conducted the data extraction and risk bias assessment

Response- Thank you. We included observational study design such as cross-sectional, case-control and cohort study. We have also added study designs to inclusion criteria in the current version of the manuscript. 

The operational definitions can be part of the introduction, however, it would be important to keep the ones that are relevant for the focus of the paper

Response- We thank you for the comment. Since the PRISMA guideline recommend separate operational definition, we put it in the method section as you recommended (keep the ones that are relevant for the focus of the paper). 

When it came to the results it was difficult to understand which outcome was being reported it would be great to analyze the different outcomes separately - also conduct a qualitative synthesis to find out what the authors mean by perinatal death (as this will include both stillbirth and neonatal death). Analyze neonatal death and stillbirth separately and if you are including stillbirth as an outcome you may have revise the population in your PICO question

Response- Thank you. We did separate analysis for stillbirth and early neonatal death as you recommended. The current PICO is more comprehensive that included both stillbirth and early neonatal death. It included fetal or newborn deaths during perinatal period both stillbirth and early neonatal deaths. The population is ‘newborn after 28 weeks’ gestation and survived seven days postpartum.’

It was great to see the Prisma flow diagram as it helped understand the selection process

Response-Thank you. The PRISMA flow diagram was shown in Fig 1.

Reviewer #3: Overall, the manuscript was well organized, adding to existing literature. The authors did a good job at providing sufficient background information on the issue at hand, however I would advise to provide more clarity when providing worldwide statistics on the effectiveness of prenatal care versus those for Ethiopia, thus allowing for the discussion and conclusions of the manuscript to have a stronger impact on its readers. 

Response- Thank you. We added the effectiveness of ANC globally ‘ANC is relevant intervention for successful maternal and child health, globally’. The available information on effectiveness of ANC are studies with varying or inconsistent information on the area which mandate further research or pooling the existing studies. The pooled effect of ANC on neonatal mortality worldwide and in sub-Saharan Africa showed positive association as we mentioned in the introduction section. But, we as far as our knowledge is concerned there were no pooled estimates results showing effect of ANC on perinatal outcome in sub-Saharan Africa and Ethiopia. 

I feel the main title of the manuscript and then those for the tables and figures could have been stronger, though current titles do make sense.

Response- Thank you for the comments. The title of the manuscript decided by the review authors after thorough discussion. Moreover, it has been registered on ‘PROSPERO’ as “The effect of ANC on perinatal outcome in Ethiopia” which best describe our review title. We afraid if we modify the title of the review it may not obey the registration principles and overlap with other review title. But, the title of the tables and figures were modified as you recommended it. 

 I do not have the appropriate knowledge base to determine whether the methods and statistical analyses used were most accurate for this type of research, however, the data provided in the Methods and Results sections matched the tables and figures provided. I caution authors to proofread carefully, paying close attention to capitalization and punctuation.

Response- Okay, thank you. Native Language speaker has seen the manuscript and we hope the capitalization and punctuation problems are now corrected.

---

## [Decision Letter · Decision Letter 1]

30 Oct 2020

PONE-D-20-22748R1

The effect of antenatal care on perinatal outcomes in Ethiopia: A systematic review and meta-analysis

PLOS ONE

Dear Dr. Kasiye Shiferaw,

Thank you for submitting your manuscript to PLOS ONE. After careful consideration, we feel that it has merit but does not fully meet PLOS ONE’s publication criteria as it currently stands. Therefore, we invite you to submit a revised version of the manuscript that addresses the points raised during the review process.

Please pay particular attention to the comment by teh reviwer: " the authors continue to conflate perinatal mortality, neonatal mortality and stillbirths. A stillbirth is not a live birth. This remains a major flaw in the manuscript and should be corrected."

We look forward to receiving your revised manuscript.

Kind regards,

Georg M. Schmölzer

Academic Editor

PLOS ONE

Reviewers' comments:

Reviewer's Responses to Questions

**Comments to the Author**

1. If the authors have adequately addressed your comments raised in a previous round of review and you feel that this manuscript is now acceptable for publication, you may indicate that here to bypass the “Comments to the Author” section, enter your conflict of interest statement in the “Confidential to Editor” section, and submit your "Accept" recommendation.

Reviewer #1: (No Response)

2. Is the manuscript technically sound, and do the data support the conclusions?

Reviewer #1: Yes

3. Has the statistical analysis been performed appropriately and rigorously? 

Reviewer #1: Yes

4. Have the authors made all data underlying the findings in their manuscript fully available?

Reviewer #1: Yes

5. Is the manuscript presented in an intelligible fashion and written in standard English?

Reviewer #1: Yes

6. Review Comments to the Author

Reviewer #1: The authors of this manuscript have addressed many of the comments satisfactorily and overall the manuscript has a nice flow with improved clarity of writing. However, there are still syntax, punctuation and spelling errors (minor concern that is easily corrected), but more importantly, the authors continue to conflate perinatal mortality, neonatal mortality and stillbirths. A stillbirth is not a live birth. This remains a major flaw in the manuscript and should be corrected.

Of note - WHO and UNICEF The Every Newborn Action Plan: Goals for reducing newborn mortality and preventing stillbirths:

Goal 1. End preventable newborn deaths.

By 2030, all countries will have reached the target of 12 or less newborn deaths per 1000 live births and will continue to reduce death and disability, ensuring that no newborn is left behind.

Goal 2. Ending preventable stillbirths

By 2030, all countries will have reached the target of 12 or fewer stillbirths per 1000 total births and will continue to improve equity.

OR (updated),

Goal 1. End preventable newborn deaths.

By 2035, all countries will reach the target of 10 or less newborn deaths per 1000 live births and continue to reduce death and disability, ensuring that no newborn is left behind.

Goal 2. Ending preventable stillbirths

By 2035, all countries will reach the target of 10 or less stillbirths per 1000 total births and continue to close equity gaps.

Specific comments:

1. In the Introduction section: Lines 68-69 and line 228. It would be helpful for international readers if the “focused ANC model” referenced a few times in the manuscript is explained.

2. In the Introduction section: Line 70. “Reduction in the gaps reductions” – needs to be clarified.

3. In the Methods section: Line 129. Authors appear to suggest that an attempt was made to obtain missing data from its source, but not always successful – authors should add a statement as to how missing data was handled in the analysis.

4. In the Methods section: Lines 174-178. Authors should separate newborn deaths and stillbirths. Hence, clarify and probably show numbers used as numerator and denominator for each of the following: Stillbirth rate per 1000 total births = number of stillbirths/total number of births, whereas neonatal mortality rate per 1000 live births = number of neonatal deaths/total number of live births. As currently stated in the manuscript, there is an unresolved discrepancy in reporting. See also Line 231.

5. In the Results section: Line 248-249. “…58% and 66% lower risk of perinatal mortality and stillbirth among women who attended at least one ANC..” – these percentages may need to be modified after recommendations in Comment #4.

7. PLOS authors have the option to publish the peer review history of their article (what does this mean?). If published, this will include your full peer review and any attached files.

Reviewer #1: **Yes: **Janine Y Khan

---

## [Author Response · Author response to Decision Letter 1]

17 Dec 2020

Response to reviewers 

Comments 

Reviewer #1 

The authors of this manuscript have addressed many of the comments satisfactorily and overall the manuscript has a nice flow with improved clarity of writing. However, there are still syntax, punctuation and spelling errors (minor concern that is easily corrected), 

Response- The comments were corrected in the main document. Thank you. 

but more importantly, the authors continue to conflate perinatal mortality, neonatal mortality and stillbirths. 

Response- We found your comment astonishing. We learnt a lot from you frankly. We defined these terms in the operational definition part. As you said these terms are not the same. We cautiously analyzed these findings in current version of the manuscript. We have explained this concern in detail below on response we provided to specific comment #4.

Of note - WHO and UNICEF The Every Newborn Action Plan: Goals for reducing newborn mortality and preventing stillbirths:

Goal 1. End preventable newborn deaths.

By 2030, all countries will have reached the target of 12 or less newborn deaths per 1000 live births and will continue to reduce death and disability, ensuring that no newborn is left behind.

Goal 2. Ending preventable stillbirths

By 2030, all countries will have reached the target of 12 or fewer stillbirths per 1000 total births and will continue to improve equity. OR (updated),

Goal 1. End preventable newborn deaths.

By 2035, all countries will reach the target of 10 or less newborn deaths per 1000 live births and continue to reduce death and disability, ensuring that no newborn is left behind.

Goal 2. Ending preventable stillbirths

By 2035, all countries will reach the target of 10 or less stillbirths per 1000 total births and continue to close equity gaps. 

Response- Thank you, it is nice recommendation. We took goal 1 End preventable newborn deaths.

By 2030, all countries will have reached the target of 12 or less newborn deaths per 1000 live births and will continue to reduce death and disability, ensuring that no newborn is left behind in our manuscript as we explained it in the current version. We agreed all goals are reducing the newborn death and making sure all countries are achieving the goal. 

Specific comments:

1. In the Introduction section: Lines 68-69 and line 228. It would be helpful for international readers if the “focused ANC model” referenced a few times in the manuscript is explained.

Response- Thank you for the comment. We defined it as ‘Focused ANC model is four visits providing essential evidence based interventions – a package to achieve the full life-saving potential that ANC promises for women and babies.’

2. In the Introduction section: Line 70. “Reduction in the gaps reductions” – needs to be clarified. 

Response- The comment was accepted. Thank you. The repeated phrase was removed in the new version of the manuscript. We rewrote the sentence. 

3. In the Methods section: Line 129. Authors appear to suggest that an attempt was made to obtain missing data from its source, but not always successful – authors should add a statement as to how missing data was handled in the analysis. 

Response- As you know missing data is challenging most of the time. We tried to request for further information when we thought the data lack clarity and/or missing and unfortunately there was no study which was excluded due to clarity and/or missing data.

4. In the Methods section: Lines 174-178. Authors should separate newborn deaths and stillbirths. Hence, clarify and probably show numbers used as numerator and denominator for each of the following: Stillbirth rate per 1000 total births = number of stillbirths/total number of births, whereas neonatal mortality rate per 1000 live births = number of neonatal deaths/total number of live births. As currently stated in the manuscript, there is an unresolved discrepancy in reporting. See also Line 231. 

Response- Thank you for the comments. Stillbirth rate is calculated using numerator of number of stillbirths and denominators of number of births (dead or alive) whereas early neonatal mortality is calculated using numerator of neonatal death and denominator of live birth during 7 days of life. 

Number of stillbirths per 1000 births (live and stillbirths). It was 38 per 1000 total births (stillbirths and live births) in our review.

Early neonatal mortality rate is the death of neonate per 1000 live births within 7 days of life. The rate was 19 per 1000 live birth after correction was made based on your recommendation. 

Similarly, the denominator used for perinatal mortality rate (PMR) determination was total births after 28 weeks of gestation. Since the majority of the studies included reported PMR as per 1000 total births, data presented in some primary studies with a denominator of total live births were changed into total births by determining new PMR taking into account the reported total deliveries, total still births and total early neonatal deaths. Therefore, the perinatal mortality rate in this finding was 41 per 1000 total births (total deliveries, total still births and total early neonatal deaths) as we mentioned in the previous version. 

5. In the Results section: Line 248-249. “…58% and 66% lower risk of perinatal mortality and stillbirth among women who attended at least one ANC..” – these percentages may need to be modified after recommendations in Comment #4. 

Response- Thank you for the comments. There is no any relationship between these percentages and definitions of these terms (comments #4). These percentages were taken from risk ratio which were found from random effect model calculation. Hence the risk ratio for perinatal mortality and stillbirths were 0.42 and 0.34 making 58% (1-0.42) and 66% (1-0.34) percentages. We did this because experts recommend explaining finding in percentage, rather putting as obsolete number.

---

## [Editor Report · Decision Letter 2]

21 Dec 2020

The effect of antenatal care on perinatal outcomes in Ethiopia: A systematic review and meta-analysis

PONE-D-20-22748R2

Dear Dr. Kasiye Shiferaw,

We’re pleased to inform you that your manuscript has been judged scientifically suitable for publication and will be formally accepted for publication once it meets all outstanding technical requirements.

Kind regards,

Georg M. Schmölzer

Academic Editor

PLOS ONE
---

## [Editor Report · Acceptance letter]

28 Dec 2020

PONE-D-20-22748R2 

The effect of antenatal care on perinatal outcomes in Ethiopia: A systematic review and meta-analysis 

Dear Dr. Shiferaw:

I'm pleased to inform you that your manuscript has been deemed suitable for publication in PLOS ONE. Congratulations! Your manuscript is now with our production department. 

Kind regards, 

on behalf of

Dr. Georg M. Schmölzer 

Academic Editor

PLOS ONE